# Simplified sewerage to prevent urban leptospirosis transmission: a cluster non-randomised controlled trial protocol in disadvantaged urban communities of Salvador, Brazil

Cleber Cremonese [1], Fabio Neves Souza [1,2,3]
Fabiana Almerinda Gonçalves Palma,[1,2] Jonatas Fernandes Araújo Sodré,[1]
Ricardo Lustosa Brito,[1] Priscyla dos Santos Ribeiro [3,4,5]
Juliet Oliveira Santana,[2,6] Rachel Helena Coelho,[7] Juan P Aguilar Ticona [1,2]
Romero J Nazaré,[2] Daiana de Oliveira,[1,2] Cainã Queiroz Silva,[1] Max T Eyre,[8]
Vinícius de Araújo Mendes,[7] Jackie Knee [8] Paula Ristow,[2,3,5]
Christine E Stauber,[9] Yeimi Alexandra Alzate López,[1] Emanuele Giorgi,[10]
Peter J Diggle [10] Mitermayer Galvão G Reis,[2,11,12] Oliver Cumming [8]
Albert Ko,[2,12] Federico Costa[1,2,10,12]

CC, FNS and FAGP contributed equally.

For numbered affiliations see end of article.

**Correspondence to**
Dr Cleber Cremonese;
cleber.cremonese@gmail.com

## ABSTRACT

**Introduction** Leptospirosis is a globally distributed zoonotic and environmentally mediated disease that has emerged as a major health problem in urban slums in developing countries. Its aetiological agent is bacteria of the genus *Leptospira*, which are mainly spread in the urine of infected rodents, especially in an environment where adequate sanitation facilities are lacking, and it is known that open sewers are key transmission sources of the disease. Therefore, we aim to evaluate the effectiveness of a simplified sewerage intervention in reducing the risk of exposure to contaminated environments and *Leptospira* infection and to characterise the transmission mechanisms involved.

**Methods and analysis** This matched quasi-experimental study design using non-randomised intervention and control clusters was designed to assess the effectiveness of an urban simplified sewerage intervention in the low-income communities of Salvador, Brazil. The intervention consists of household-level piped sewerage connections and community engagement and public involvement activities. A cohort of 1400 adult participants will be recruited and grouped into eight clusters consisting of four matched intervention-control pairs with approximately 175 individuals in each cluster in baseline. The primary outcome is the seroincidence of *Leptospira* infection assessed through five serological measurements: one preintervention (baseline) and four postintervention. As a secondary outcome, we will assess *Leptospira* load in soil, before and after the intervention. We will also assess *Leptospira* exposures before and after the intervention, through transmission modelling, accounting for residents' movement, contact with flooding, contaminated soil and water, and rat infestation, to examine whether and how routes of exposure for *Leptospira* change following the introduction of sanitation.

**Ethics and dissemination** This study protocol has been reviewed and approved by the ethics boards at the Federal University of Bahia and the Brazilian National Research Ethics Committee. Results will be disseminated through peer-reviewed publications and presentations to implementers, researchers and participating communities.

**Trial registration number** Brazilian Clinical Trials Registry (RBR-8cjjpgm).

## STRENGTHS AND LIMITATIONS OF THIS STUDY

⇒ The first rigorous controlled intervention study for the effect of sanitation on *Leptospira* transmission in disadvantaged urban communities.

⇒ This is a cluster-based non-randomised controlled trial with intervention clusters receiving a simplified sanitation intervention and matched control clusters no intervention.

⇒ The alliance between the natural and social sciences and the potential that it has to deepen community perceptions about leptospirosis and interventions before and after, as well as involvement in it in some decision-making processes in the improvement of their health, well-being and sanitation.

⇒ The allocation is non-randomised with selection based on intervention criteria determined by the implementer may have diluted intervention effects.

⇒ Intervention will be carried out by the local public water and sanitation company, following their own planning, with no input or interference from the research team in the schedule or implementation.

## INTRODUCTION

Globally, leptospirosis is a leading zoonotic disease in terms of morbidity and mortality,[1] causing >1 million cases and >50 000 deaths yearly.[1–3] This zoonosis is an environmental and sanitation-related disease[4] characterised by annual epidemics during periods of increased rainfall and flooding.[5 6] Infection occurs frequently following exposure to water and soil that has been contaminated with the urine of infected animals.[3 7] Leptospirosis is an emerging disease in urban areas of most low-income and middle-income countries,[1] where a significant fraction of the population lives in disadvantaged urban communities (often called slums or *favelas* in Brazil).[8] In those settings, poverty, precarious housing conditions, overpopulation and lack of basic sanitation enable a high density of rats (the main urban reservoir hosts[9–12]) and high leptospirosis transmission.[13 14] As the population living in urban disadvantaged communities will double to 2 billion by 2025[15] the burden of leptospirosis is expected to increase.

Public health intervention strategies to prevent leptospirosis transmission are essential and must target the infrastructure deficiencies to prevent or diminish human exposure to environmental contamination sources. Governmental interventions focused on improving sanitation are critical for reducing the burden of leptospirosis as well as other diseases, like diarrhoeal.[4 16] However, these interventions, especially in disadvantaged urban communities, are not a priority or foreseeable soon considering the high cost of providing and maintaining such sanitation infrastructure. Exclusion of the disadvantaged urban communities' health issues in the government's political agenda,[17–19] contributes to historical and structural disparities. Brazil, since the 80s, has been recognised as a global leader in the development of cost-effective water and sanitation systems for densely populated urban settings.[20 21] These systems, called simplified or 'condominial' water and sewerage systems, are designed to be installed in densely populated, informal communities that are often situated in the kind of hilly terrain, common in Brazilian *favela* communities. The simplified sewerage approach is unique because it redefines the service unit as a community block rather than an individual household. It uses smaller pipes and inspection chambers (or cleanouts) at key junctions to facilitate installation and maintenance.[20 21] The system design reduces costs by decreasing the length of the trenches needed for sewerage lines and by using smaller pipes made from less expensive materials such as polyvinyl chloride. The installation of this alternative system can cost 50%–70% less than a conventional system.[20] The lower costs associated with the installation of simplified systems makes these systems a more feasible alternative to conventional sewer systems in poor, resource-constrained settings. However, the simplified sewerage approach requires intensive community mobilisation and engagement[22] as it is residents who take on leadership roles and have responsibility for decision-making.

Health impact evaluations of simplified sewerage approach in preventing disease has previously been limited to diarrhoeal diseases.[23] The evaluation of such interventions is challenging because of methodological problems such as cost and time of implementation, the requirement of large sample sizes, difficulties with randomisation of the intervention allocation, and confounding factors (eg, rodent infestation and social-economic status). Contact with environments contaminated with sewer overflow or residents' exposure to open sewerage near the household can influence the degree of individual exposure to *Leptospira* bacteria.[13] Humans exposure to *Leptospira* contamination through poor sanitation is mediated by individual behaviour, socioeconomic and environmental factors, rainfall and flooding.[4] Therefore, the mechanisms of *Leptospira* transmission have different but important components of the environmental transmission route, for which their relative contribution is unknown because of confounding factors. Only studies that integrate a rigorous characterisation of contamination sources (*Leptospira* load) and residents' movement could separate their effects and understand how source reduction interventions impact those transmission mechanisms. Determining the contribution of each of these mechanisms to *Leptospira* infection would help us to simultaneously understand the relationships and pathways that drive transmission.[4] This information is key to making interventions generalizable to other environmentally transmitted diseases that affect urban disadvantaged populations.

In this protocol, we describe an evaluation of a simplified sewerage intervention and community engagement with a quasi-experimental study design using non-randomised intervention and control clusters. We aim to assess the longitudinal impact of a simplified sewerage intervention on the incidence and transmission mechanisms of *Leptospira* infection. In addition, we intend to investigate the impact of these interventions on reducing other water and environmental-related transmissible diseases.

### Research aims and hypotheses

We hypothesise that the combination of a simplified sewerage infrastructure and community engagement activities will reduce the risk of *Leptospira* infection among disadvantaged urban communities in Salvador, Brazil. We further hypothesise that this approach will be effective in reducing: (1) frequency of human contact with open sewerage contaminated with pathogenic *Leptospira*; (2) rat infestation and (3) *Leptospira* load in the soil around the simplified sewerage. We aim to evaluate the efficacy of a sewerage intervention on reducing *Leptospira* infection and other health outcomes related to sanitation in disadvantaged urban communities in Brazil. Our specific objectives are to: (1) to evaluate prospectively the effectiveness of a simplified sewerage intervention in reducing the risk of *Leptospira* infection; (2) to determine the mechanisms by which the simplified sewerage reduce direct human contact with environmental pathogen load in urban

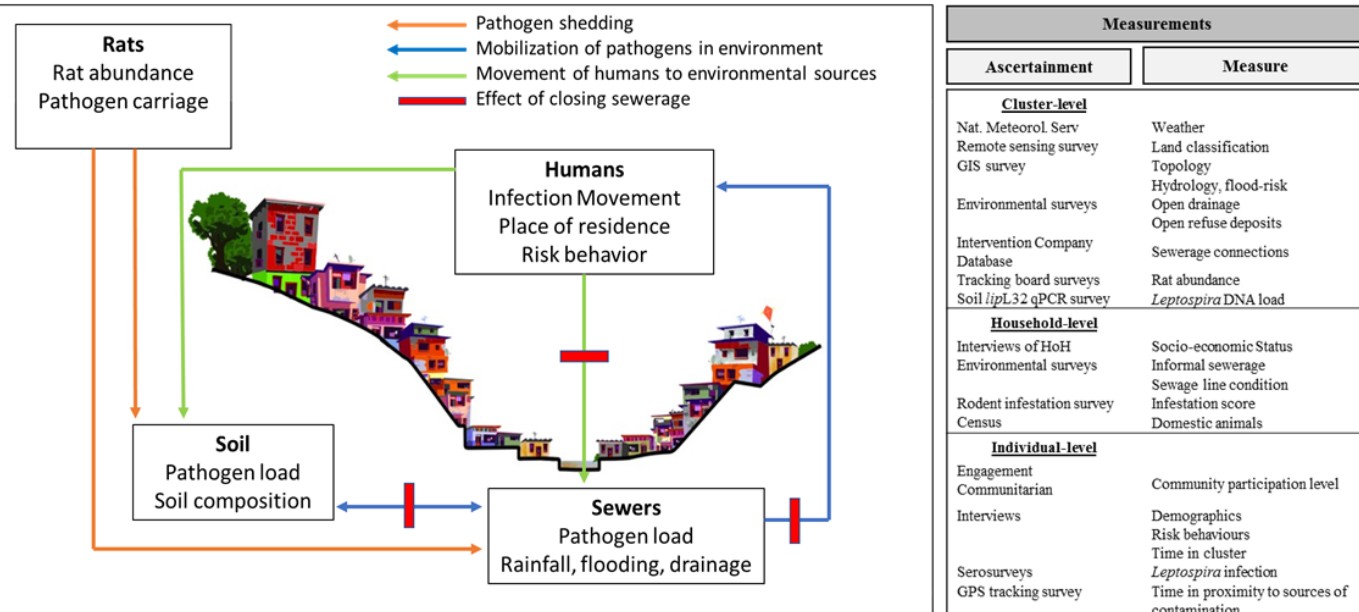

**Figure 1** Rationale, hypothesis and outcomes assessed in the study. We hypothesise that sanitation intervention will improve the environmental conditions and change the behaviour of humans, resulting in both direct and indirect impacts on the incidence of leptospirosis and other diseases human and well-being. HoH, household-level; qPCR, quantitative PCR.

slums. Figure 1 provides further details on our hypotheses concerning the effects of the sanitation intervention on the transmission mechanisms of urban leptospirosis. We conceptualise this protocol as focusing on the sanitation intervention to decrease leptospirosis; however, this study protocol generates rich opportunities to evaluate other health outcomes. Therefore, we additionally aim to investigate the effects of the sewerage intervention on childhood exposure to a range of enteropathogens, and long-term child neurodevelopment and anthropometry. As exploratory outcomes, we will also investigate the effect of the sewerage intervention on the reduction of risk of arbovirus infectious diseases.

## METHODS AND ANALYSIS
### Overview of study design
This is an intervention trial with a cluster non-randomised controlled trial, with eight clusters (four intervention and four control), including longitudinal serological follow-up of 1400 adults' residents from disadvantaged urban communities at high risk for leptospirosis transmission in Salvador, Bahia, Brazil. The primary outcome will be *Leptospira* infection. Each pair of intervention and control clusters will be allocated in distinct hydrological sub-basins with distances from 1000 to 1500 m apart. Clusters will be matched on socioeconomic status (SES), topographic or hydrological characteristics, population density, and *Leptospira* seroprevalence. The cluster selection will also consider exposure to the pathogen by selecting houses that are located to similar distances to new sewerage system and control open sewers. The intervention aims at a combination of sanitation (simplified

sewerage intervention) and community engagement activities.

### Study settings
Salvador is the third largest city in Brazil with a population of 2.9 million residents.[24] Using a surveillance database of active leptospirosis in the study setting between 1996 and 2018, we performed a population-based, adjusted Kernel analysis among geo-localised records of households with leptospirosis cases to identify high-risk area corresponding to 8.06% of the total area of the city.[14 25] We selected eight clusters (four intervention and four control) with average size 0.03 km² located in these areas. The intervention clusters will be located in areas where the provision of simplified sewerage and community engagement is planned. Moreover, the choice of the control group is dependent on the implementer's information regarding non-intervention planned in the area for a short-term period of 5 years. Geographic location of intervention and control clusters is detailed in figure 2.

### Description of the sewerage intervention
During 2022 and 2023, the sewerage intervention (simplified sewerage intervention) will be implemented by the Public Water and Sanitation Bahia State Company (*Empresa Baiana de Águas e Saneamento*) in different urban locations in Salvador, Brazil. The expansion of the sanitation coverage in the city has generated an opportunity to assess the longitudinal effectiveness of the intervention on the transmission of sanitation-related diseases, such as leptospirosis. The company is government-owned, with responsibilities for managing, monitoring and distributing water and sanitation services throughout the state of Bahia. The costs, management and entire execution of

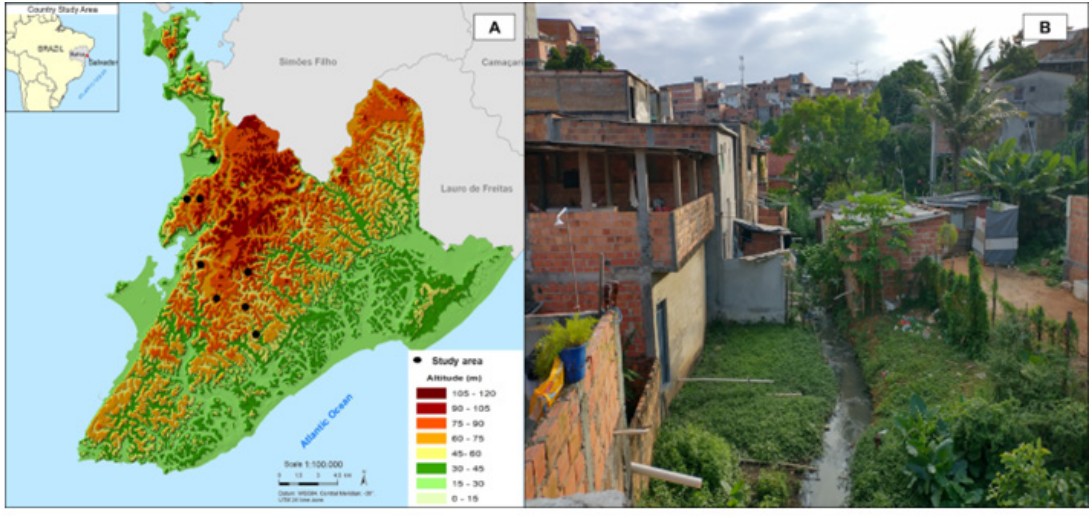

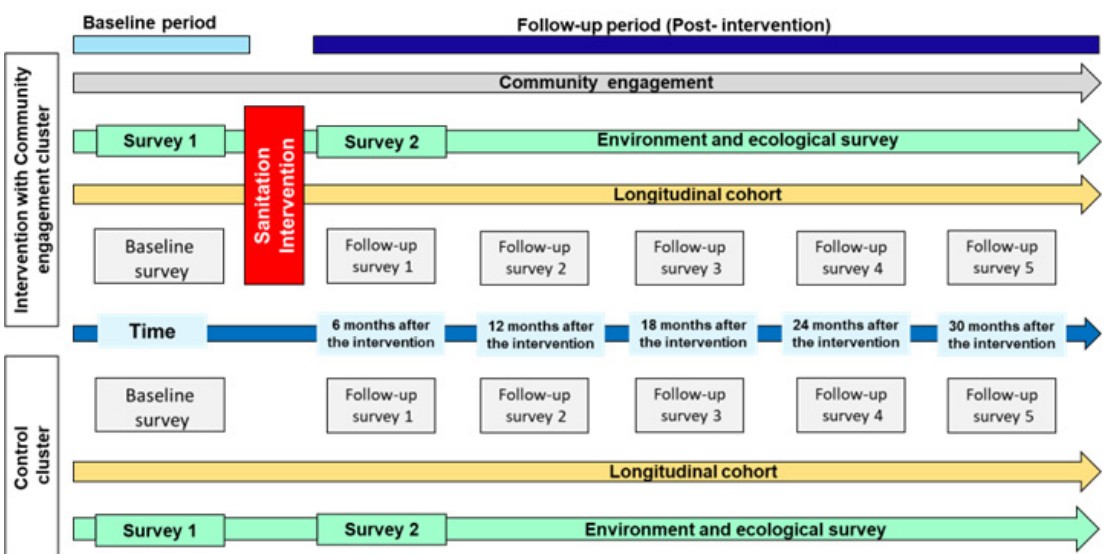

**Figure 2** Study clusters in the city of Salvador, Brazil. Altitude and sampling clusters (A). Example of baseline sanitation conditions of the study cluster (B). In (C) consort flow diagram by intervention and control cluster, data collection times and type of activities.

the work will be the responsibility of the water and sanitation company, while the team of researchers will only monitor the development of the work.

This simplified sewerage intervention is characterised by receiving the sewage generated by a cluster of houses instead of individual houses. In conventional sewerage system designs, each household is connected directly to a principal line while in the sewer simplified system households are connected to branch lines within each block[23] (online supplemental figure 1).

The sanitation company will carry out the sewerage network works, which include the installation of a 150 mm primary pipe at the bottom of the residential lot, bordering a watercourse. Household sewerage networks are interconnected on one main line connecting to the main network, thus the intervention is expected to increase sanitation coverage in the study clusters by expanding the collection network. These interventions would exclusively be of a sewerage nature, with no complementary functions such as local drainage. Importantly, the intervention also includes community participation in the action designs and implementation.

### Community engagement

Simplified sewerage includes different degrees of social participation. We recognise that the intervention in sanitary sewerage impacts the local environment and the communities' perception is affected by the sanitation interventions. Problems and failures related to sewerage intervention have been reported as mainly due to the difficulties in the relationship between the technicians who accompany the works and the communities

which reduced the community participation to sewerage maintenance.[22]

Faced with this complexity related to urban sanitation, the community engagement processes that will be implemented in this study are focused on approaches that would stimulate community mobilisation to participate in the research processes, popular health education (PHE) actions, and in the evaluation of the implementation of sewerage intervention. PHE actions will be oriented toward disease prevention and health promotion of the population. The PHE is a 'set of practices targeted at individuals and the community',[26 27] and this study intends to contribute to community empowerment and support for the management of sewerage intervention. The community engagement activities would be implemented in stages and with different stakeholders, including the community leaders, residents, and a multidisciplinary team of researchers including epidemiologists, ecologists, sanitation engineers, anthropologists, environmental microbiologists, geographers, economists, statisticians, etc. The study, therefore, aims to strengthen community participation in the decision-making in the management of sewerage intervention, and also support the involvement of residents in activities research such as (1) community and institutional mobilisation; (2) collaborative mapping and (3) PHE actions. Moreover, we will perform community ethnography to deepen the understanding of the social and cultural impacts caused before and after the intervention, as well as its environmental impacts and the mobilisation processes.

### Study population and sample size

To evaluate the effectiveness of the simplified sewerage intervention in reducing the risk of *Leptospira* infection, we plan to recruit an estimated 1400 participants aged ≥18 years, residing in the selected clusters (approximately 175 adults per cluster in baseline). To evaluate the intervention effect on secondary (sewerage contact, environmental pathogen load, long-term child development) and exploratory (arbovirus infections, diarrhoeal disease and enteric infection) outcomes, we plan to increase our sample size by 420 (30%) participants, aged between 6 months and 17 years, totalling 1840 people. We considered a potential loss to follow-up rate of 15%–20% per year. Therefore, by including additional participants, we ensure the maintenance of an appropriate sample size for the outcomes that will be studied. All participants from both intervention and control clusters will be followed over a period of 3 years after the sewerage intervention and a baseline prior to the intervention (figure 2). In the flow diagram, we provide a timeline and trial study phases in detail (figure 3).

The sample size was estimated using data on *Leptospira* infection from a previous leptospirosis cohort study in Salvador, Brazil.[13] We divided the previous leptospirosis cohort study[13] into areas of similar size to that the clusters used trial and calculated the between-cluster variation. Of 1840 individuals participating in the previous cohort,

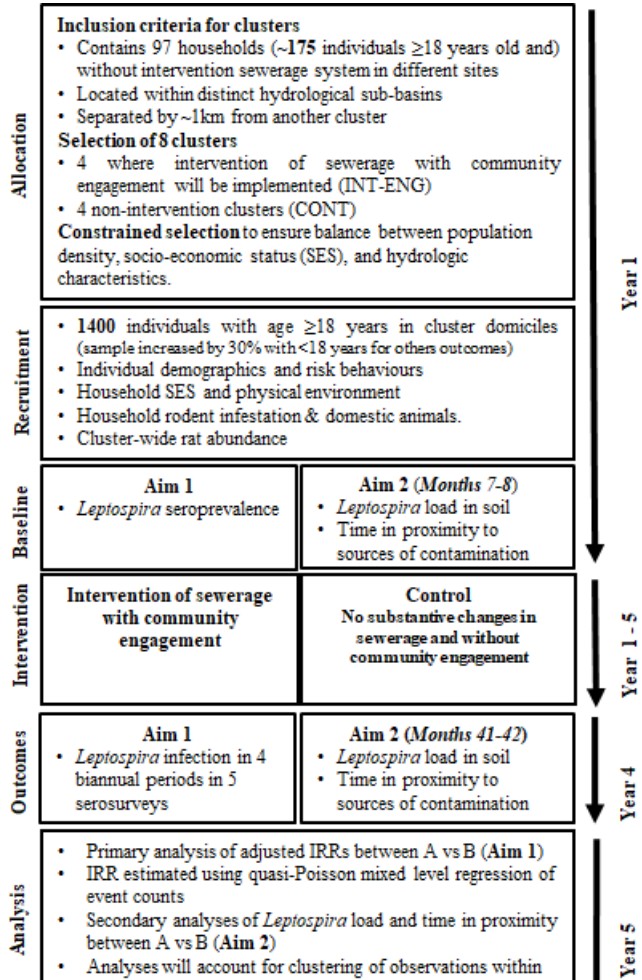

**Figure 3** Overview of the study procedures, the timeline for study and data collection. Note 1: Each cluster of the study will be formed by a cluster with intervention versus control. The period of activities for each serological survey will take place at the time described in the figure (allocation, recruitment and baseline). Note 2: Update years 1 to 5 will be realised after receipt of the water and sanitation company schedule.

142 (7.9%) experienced leptospiral infection (k/Intracluster Correlation Coefficient (ICC)=0.03). Power calculations are based on a cluster-level using effect sizes (% decrease in incidence between intervention and control clusters) 40%, 50% and 60%, assuming an infection rate of 0.08 per person-year. All sample size calculations used methods for matched cluster-randomised trials with the study structure of four matched pairs of clusters. Sample size estimates assumed a 12% annual drop-out rate across the 3-year study period.

### Control selection

Control clusters (CONT) will be selected in locations with an incomplete sewerage sanitation system. Matching criteria for control clusters include: location in a geographic area that has similar environmental (elevation, hydrological data) and social (income, education, population density) characteristics to the intervention

clusters; and at a distance of 1000 and 1500 m from the intervention cluster. Areas that received major infrastructure interventions and sanitation improvements in the last 5 years will be excluded from the selection frame for control clusters.

### Eligibility and enrolment

Before the recruitment of study participants, a census will be carried out within the selected eight clusters to enumerate all residents from households within <40 m distance from an open sewer. This would ensure that households and residents living in areas with a higher risk of leptospirosis transmission are included. At each census, the individual data (name, sex, age) and the geo-coordinates a primary household respondent would be collected. As a further step, we will randomly select population-based cohorts with 95 households (175 participants) per cluster. Residents will be eligible for inclusion in the study, if (1) they are 18 years of age or above, (2) have lived in the area for at least 6 months, (3) sleep in the household for at least three nights per week and (4) consent to the study procedures and complete an informed consent form. However, residents will be excluded if: (1) they do not meet all of the inclusion criteria; (2) they cannot provide written informed consent to participate in the study, (3) they refuse or are unable to provide demographic information and biological sample; (4) they are with limited mental health problems or other problems that have implications for survey responses or participation in any other phase of research activity.

For secondary outcomes we will include all children and young between 6 months and 17 years of age will be included in the study (online supplemental table 1). We will apply similar inclusion and exclusion criteria as described but excluding children for which their legal guardians do not provide written informed consent for their participation.

### Outcomes
#### Primary outcome measure

The study trial will consist of three specific phases: (1) baseline; (2) sanitation intervention implementation; (3) follow-up of the residents and evaluation of the sanitation intervention (figure 2). During the phase (1) a baseline survey will be conducted to characterise the demographics, SES and *Leptospira* seroprevalence in each cluster. During the phase (3) biannual serosurveys in the cohorts will be conducted to measure the primary outcome, *Leptospira* asymptomatic infection, and estimate the risk of infection during the dry and rainy seasons. The field team will collect blood samples from eligible participants with age ≥18 years. A blood sample (10 mL) will be collected from each study participant and examined for *Leptospira* antigen using the microscopic agglutination test diagnostic (MAT).[4] MAT is the standard test to determine *Leptospira* infection in longitudinal studies and *Leptospira* infection will be defined as a fourfold rise in MAT titre or seroconversion (negative to ≥1:50) between samples from

consecutive serosurveys. We will use a panel of six reference strains: representing *L. interrogans* serovars Autumnalis strain Akiyami A, Canicola strain H. Utrecht IV and Icterohaemorrhagiae strain RGA, *L. borgpetersenii* serovar Ballum strain MUS 127, *L. kirschneri* serovar Cynopteri strain 3522C and Grippothyphosa strain Duysterand the two clinical isolates: *L. interrogans* serovar Copenhageni strain Fiocruz L1-130 (isolated locally in 1996) and *L. santarosai* serogroup Shermani strain LV3954 (isolated locally). This procedure had the same performance in identifying MAT seroconversion in our prospective studies[4 14] compared with the WHO recommended panel of 19 reference serovars. The screening will be performed with serum dilutions of 1:50–1:100. When agglutination is observed at 1:100, the sample will be titrated to determine the highest titre. As part of quality control procedures, two independent evaluations will be conducted for 8% of the samples and all samples from infected subjects.

### Secondary outcomes
#### *Environmental exposure secondary outcomes*

Secondary outcomes for environmental effects will include microbiological assays to assess soil contamination with pathogenic *Leptospira*. The presence and concentration of pathogenic *Leptospira* will be evaluated in the eight clusters. Bases on prevalence the contamination soil with *Leptospira* in studies previous and packing density of 0.4 with minimum distances between collection points, a total of 68 soil samples and 15 water sewage samples will be collected in each cluster and transported refrigerated to the laboratory within 6 hours for processing. Sampling will take place in duplicate, with intervals of 7–9 days between them. After processing, soil and sewage samples will be stored at −80°C until molecular assays can be performed. The presence and concentration of pathogenic *Leptospira* will be evaluated using a qPCR assay (quantitative PCR), targeting the *lipL32* gene. Samples will be considered positive when the two qPCR replicates show amplification before a Cq of 40.[7] Environmental variables for rainfall, temperature, landscape (remote sensing using drones), flooding and hydrological features will also be collected during rainy seasons before and at least 3 months immediately after the sewerage intervention.

#### *Rats survey*

Rat abundance will be examined at each cluster using tracking boards to quantify the level of rat activity, before and after the interventions. The use of tracking boards to quantify rat presence and activity (as rat abundance proxy) will be performed, following previously described, validated and optimised methods.[28] Tracking boards will be distributed in each cluster for 2 days in a row and will be scored daily. The overall level of rat presence for each cluster will be estimated based on the percentage of positive tracking boards and the intensity of marks on each board before and at least 3 months immediately after the sewerage intervention. We will compare the abundance

and activity of rodents with the environmental leptospiral load in the area.

## Human movement

Environmental exposure to *Leptospira* (ie, proximity to open sewer and unpaved roads; housing conditions; and flooding) will be quantified by studying the individual movement of the participants in each study cluster. Movement evaluations will be performed biannually, before and after the sewerage intervention. We will focus on a subcohort of 320 residents living in each of the eight study clusters, divided into four groups: young males (18–34 years), young females (18–34 years), old males (aged ≥35 years) and old females (aged ≥35 years). We will use GPS data loggers that can measure small-scale movement patterns.[29 30]

Forty residents in each of the study clusters will be invited to use a GPS data logger IgotU GT-600 device (Mobile Action Technology, New Taipei City, Taiwan).[31] Additionally, we will obtain information about movement and behaviour through resident diaries and retrospective mapping before and after the intervention. Diaries will provide qualitative information on the number of hours inside/outside the study cluster and the household; contact duration and nature of the contact (ie, contact duration with and without use of boots) with contaminated sources; and the activity performed (ie, working, leisure, etc) for 7 days. Retrospective mapping will provide information on the past 24 hours' pathways travelled by the residents, and contacts with contamination sources from the residents' perspective.[30] To determine participants' compliance with the study protocol using the GPS data logger and diary entry is defined as adherence for blood collection and direct observations will be conducted for participants ≥18 years of age. Noncompliance is defined as non-adherence to all study steps.

## Other outcomes

### Human health and well-being

The data will be collected preintervention and postintervention to evaluate the impact on well-being between arms. Other outcomes related to individual and household health and well-being include subjective measures—selfreported general, physical and emotional well-being using the Short-Form Health Survey-12 (SF-12);[32] perceived living conditions questions; the short version (five items) of the Brazilian Food Insecurity Scale[33] which measures the perception and experience of hunger over the past 3 months; and objectives measures—productivity through employment and income. We will also evaluate whether there will be a decrease in material and economic loss measures due to floods and landslides—which often occur in the areas without adequate sanitation.

### Long-term child development

Enteric pathogens associated with diarrhoea can inhibit a child's ability to absorb nutrients, further exacerbating malnutrition and developmental delays. Studies are

indicating a growing association between enteric infections and environmental enteropathy, a disorder characterised by reduced intestinal barrier function, abnormal intestinal morphology and increased inflammation.[34 35] This disorder is thought to stem from unsanitary conditions that lead to repeated exposures to enteric pathogens. Additionally, poor gut health, chronic immune stimulation and malnutrition, are linked to poor neurodevelopment in children, as well as stunting. In poverty-stricken environments, these nutritional and environmental risks can also affect the mother–child behaviour patterns, which are critical for normal development, including a balance between time spent with caring their children and sanitation home issues.[36]

Child neurodevelopment will be evaluated using the Ages and Stages Questionnaires third edition (ASQ-3) for children aged 6–66 months and with the Brazilian National Assessment of School Performance (ANRESC– *Avaliação Nacional do Rendimento Escolar*) for children older than 66 months, in the preintervention and postintervention period. ASQ-3 is a tool used to monitor the child's developmental progress, it includes fine and gross motor, communication, problem-solving, and personalsocial domains.[37] ANRESC is part of the Brazilian Basic Education Assessment System, which included the assessment of Math and Language skills.[38] The child development assessment will also include measurements of weight, height, body mass index and head circumference. We will use the WHO Child Growth Standards and the results will be summarised using the Z-score.[39]

### Enteric diseases

To evaluate the impact of the intervention on enteric pathogen exposure among young children and, we will collect stool and/or rectal swab samples from children aged 6 months to 5 years old in the preintervention and postintervention periods. We will use a custom-designed TaqMan Array Card (TAC) to analyse samples for the presence of over 30 enteric pathogens, including viruses, bacteria, protozoa and helminths. The TAC uses microfluidic technology to simultaneously perform 48 individual qPCR per sample and has been previously validated and used for detection enteric pathogens in stool.[40–42]

### Arboviral diseases

We will evaluate the effectiveness of the sewerage intervention and the changes in landscape characteristics associated with them, in the spatiotemporal distribution of arbovirus (Dengue, Zika, Chikungunya) vector mosquitoes. We will carry out entomological investigations in private (participants' homes) and public areas with the collection of adult and immature mosquitoes by aspiration,[43] with the verification of identified mosquitoes breeding sites, to measure their density. We will verify whether there is a spatiotemporal pattern in the presence and density of vector mosquitoes in the sampled regions, to allow us to investigate the possible factors that may be related to changes in the distribution of exposures before,

during and after the intervention process and whether these interventions have modified ecological indices for vector abundance.

## Data collection and management

Trained interviewers will administer specific questionnaires to participants through face-to-face interviews to collect general demographic information, socioeconomic data, exposure factors and individual or collective sanitation practices, and knowledge about leptospirosis and other diseases. In the case of children at home, a new questionnaire will be answered by the caregiver. Information will be collected on the child's general health status, development, and medical history and diseases (online supplemental table 2). The questionnaires and all information will be confidential and coded using individually identifiable numeric identifiers. Every effort will be made to protect the privacy of the study participants and all data collected will be treated as confidential.

All questionnaires and forms described in this protocol will be included in the Research Electronic Data Capture (REDCap) database. The REDCap web version as well as its mobile application will be used to record all data using mobile devices. A data management staff will keep track of the information in real time.

## Participants and public involvement

All study subjects will be involved in the trial, answering questionnaires, providing biological samples, and involved in engagement activities as previously described in the section on community engagement. The sanitation intervention will be conducted by a governmental agency, and the implementation of this intervention involves dialogue with the residents of the communities. However, the study design and conduct of the trial of this intervention study will be taking place independently of public involvement. After each follow-up completion, research findings will be shared and disseminated in a group's meetings or visit households with key stakeholders.

## Statistical analysis

Descriptive statistics will be employed to report the demographic, social, economic or environmental characteristics between the study clusters (intervention and control clusters). For the primary outcome, we will apply quasi-Poisson regression to estimate the incidence rate ratio of *Leptospira* infection. We will also evaluate the effect of the intervention on *Leptospira* infection risk using mixed-effects logistic regression models to account for clustering of the observations within study units and across repeated measures for individuals. We will model the overall effect of the intervention on infection risk while controlling for baseline sewerage and flooding exposure and potential confounding variables, such as age, sex, indicators of the socioeconomic level, and rainfall intensity during an annual period, which are presumably unrelated to the intervention (figure 1).

For the secondary outcome, we will conduct environmental analyses of *Leptospira* load between intervention and control clusters to compare environmental risk. To do this, we will model two leptospiral load outcomes, prevalence (binomial) and concentration (log-transformed), using generalised linear models with random effects to account for clustering within study units. The association between environmental contamination and hypothesised predictors of risk (rat abundance, human movement and ecological data) will also be explored in the model (figure 1).

## Limitations and bias

The allocation is non-randomised with selection based on intervention criteria determined by the implementer so the risk of residual confounding cannot be excluded. The choice of the control group is also dependent on the implementer's information about the non-intervention in the area in the short term (5-year period). To limit the potential for bias arising from our non-randomised design, we will select control group with as similar as possible socioeconomic and environmental characteristics to those of their matched intervention clusters.

We will perform the impact calculation through the double differentiation of the averages of the results found between the intervention and control clusters before and after the sewerage intervention. The mitigation of bias will be carried out through the control of fixed effects of time and location between different intervention and control clusters over the years of evaluation.

## Ethics and dissemination

This project was evaluated and approved by a local ethics committee (CEP/ISC/UFBA) under number CAEE 32361820.7.0000.5030, and a national research ethics committee (CONEP) linked to the Brazilian Ministry of the Health under approval number 4.235.251. The trial study protocol was registered with the Brazilian Clinical Trials Registry, RCT trial (https://ensaiosclinicos.gov.br/rg/RBR-8cjjpgm). All participants involved in the study will provide informed consent before any study activity, and the community leaders, previously identified by the team, will provide verbal consent before enrolment of the community in the trial. Participants aged 12 and 17 will sign an assent form, in addition to a participation consent form completed through their parents or legal guardians. Children under the age of 12 will have their consent signed by their parents or legal guardians only. The results of the trial will be presented at local and international meetings and submitted to peer-reviewed journals for publication. Results will also be shared directly with the participating communities. All individual information participants will be kept and published anonymously.

**Author affiliations**
[1]Institute of Collective Health, Federal University of Bahia, Salvador, Bahia, Brazil
[2]Instituto Gonçalo Moniz, Fundação Oswaldo Cruz, Ministério da Saúde, Salvador, Bahia, Brasil
[3]Institute of Biology, Federal University of Bahia, Salvador, Bahia, Brazil

4Institute of Biological Science, Federal University of Minas Gerais, Belo Horizonte, Minas Gerais, Brazil
5National Institute of Science and Technology in Interdisciplinary and Transdisciplinary Studies in Ecology and Evolution (INCT IN-TREE), Institute of Biology, Federal University of Bahia, Salvador, Bahia, Brazil
6Institute of Geosciences, Federal University of Bahia, Salvador, Bahia, Brazil
7Faculty of Economics, Federal University of Bahia, Salvador, Bahia, Brazil
8London School of Hygiene & Tropical Medicine, Faculty of Infectious Tropical Diseases, Disease Control Department, London, UK
9Department of Population Health Sciences, School of Public Health, Georgia State University, Atlanta, Georgia, USA
10Lancaster University Lancaster Medical School, Lancaster, UK
11Faculty of Medicine, Federal University of Bahia, Salvador, Bahia, Brazil
12Department of Epidemiology of Microbial Diseases, School of Public Health, Yale University, New Haven, Connecticut, USA

**Acknowledgements** The authors gratefully acknowledge critical input on the study protocol and all the study participants. We acknowledge the partner institutions for their collaboration and dedication. We thank Patricia Lustosa and Mayara Carvalho for their help with the review and discussion of this protocol, Ana Maria Silva and George G Machado for help with supplementary figure, and Hammed O Mogaji for his help with manuscript proofing.

**Contributors** FC, CC, MGGR, AK, YAAL, PJD, CES and OC contributed with project's main conceptual ideas; FC, MTE, EG and CC contributed to study design and statistical analysis plan; FNS, DdO, FAGP, CES, JFAS, PdSR, JOS, RHC, DS, RJN, RLB, VAM, JPAT, JK, PR, FC and CC developed field and laboratory protocols and implementation; FNS, FAP, CC, RHC and FC developed data collection tools; FNS, FAGP, JFA, RHC, PR, CC and FC contributed to drafting the main manuscript. All authors contributed to editing and revising the manuscript.

**Funding** This work was supported by Wellcome Trust (grant number 218987/Z/19/Z). All project costs are jointly funded by Wellcome Trust and the Department of Health and Social Assistance (DHSC), through the National Institute for Health Research (NIHR), using the UK Official Development Assistance Fund (ODA).

**Map disclaimer** The inclusion of any map (including the depiction of any boundaries therein), or of any geographic or locational reference, does not imply the expression of any opinion whatsoever on the part of BMJ concerning the legal status of any country, territory, jurisdiction or area or of its authorities. Any such expression remains solely that of the relevant source and is not endorsed by BMJ. Maps are provided without any warranty of any kind, either express or implied.

**Competing interests** None declared.

**Patient and public involvement** Patients and/or the public were not involved in the design, or conduct, or reporting, or dissemination plans of this research.

**Patient consent for publication** Not applicable.

**Provenance and peer review** Not commissioned; externally peer reviewed.

**ORCID iDs**
Cleber Cremonese http://orcid.org/0000-0003-2700-7416
Fabio Neves Souza http://orcid.org/0000-0002-3542-8918
Priscyla dos Santos Ribeiro http://orcid.org/0000-0001-7048-6999
Juan P Aguilar Ticona http://orcid.org/0000-0002-6971-2677
Jackie Knee http://orcid.org/0000-0002-0834-8488
Peter J Diggle http://orcid.org/0000-0003-3521-5020
Oliver Cumming http://orcid.org/0000-0002-5074-8709

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
