## [Reviewer comments · BMJ Open]

ARTICLE DETAILS

TITLE (PROVISIONAL)	Simplified sewerage to prevent urban leptospirosis transmission: a cluster non-randomized controlled trial protocol in disadvantaged urban communities of Salvador, Brazil
AUTHORS	Cremonese, Cleber; SOUZA, FABIO; Palma, Fabiana; Sodr�, Jonatas Fernandes; Brito, Ricardo; Ribeiro, Priscyla dos Santos; Santana, Juliet; Coelho, Rachel Helena; Ticono, Juan P. Aguilar; Nazar�, Romero; de Oliveira, Daiana; Silva, Cain�; Eyre, Max; Mendes, Vin�cius; Knee, Jackie; Ristow, Paula; Stauber, Christine; L�pez, Yeimi Alexandra; Giorgi, Emanuele; Diggle, Peter; Reis, Mitermayer Galv�o; Cumming, Oliver; Ko, Albert; Costa, Federico

VERSION 1 – REVIEW

REVIEWER	Dos Santos , Andrea Pires Purdue University, College of Veterinary Medicine
REVIEW RETURNED	24-Jun-2022

GENERAL COMMENTS	R eview Article type, protocol; manuscript ID, bmjopen-2022-065009: Simplified sewerage to prevent urban leptospirosis transmission: a cluster non-randomized controlled trial protocol in disadvantaged urban communities of Salvador, Brazil. This manuscript proposes a simplified method to reduce the exposure of humans to leptospirosis by modifying the sewage in underserved areas of Salvador, Brazil, by applying non-randomized intervention and control clusters. It also aims to characterize the mechanisms involved in Leptospirosis transmission. The manuscript is well written and objective. Most importantly, it will address a critical problem seen in underserved communities in Brazil and other low- and middle-income countries. The approach is multidisciplinary, with appropriate methods and well-defined expected outcomes. The manuscript fits the guidelines for publication as a protocol article: It is an ongoing study, dates are included in the manuscript, no results or conclusions are present in the protocol, and no flaws that would prevent a sound interpretation of the data are observed. I have a few comments for the authors and minor corrections in the text (editorial). Comments: The authors aim to characterize the mechanisms involved in Leptospirosis transmission, but the role of domestic animals is not explored fully (only census information and no leptospirosis testing). As a suggestion, I believe the inclusion of domestic animal testing (dog/cat) would provide a more one health-focused approach and will add to the characterization of transmission mechanisms of leptospirosis. My second comment is regarding the current socio-economic situation in Brazil. The authors cite the exclusion of the
--

	disadvantaged urban communities' from the government's political agenda; however, the references are from 2003 and 2006. Although this has been a historical problem in Brazil, I believe a more recent reference would be appropriate. It is also unclear if the authors refer to the three levels of government (federal/state/municipal) or if it is a general statement. Although this is not the focus of the work, a few statements that reflect the current situation would be beneficial to the global understanding of the context of the work. I understand the sanitation company is public and run by the State (Bahia); the work is planned for 2022-2023. Do the authors consider a limitation of the work having no control over the implementation of the sewage system (subjected to possible cancellations or delays, change of power/elections)? Minor/Editorial comments: Line 36: needs a comma before and. Also, the verb is missing: ", and it is known"... Line 61: In the section "STRENGTHS AND LIMITATIONS OF THIS STUDY," the authors only included the strengths and no limitations (discussed later in the manuscript) Line 103: since diarrhea is not a disease per se, I would change it to "diarrheal" diseases or similar. Line 111: I would italicize favela throughout the manuscript Line 126: e.g., (missing a comma) Line 228-229: Popular health education actions (no caps) Line 370: quantitative polymerase chain reaction (qPCR was used previously) Line 393: The word "patient" is not ideal; perhaps participants? Socio-economic and socioeconomic are used in the text; please, standardize them to one format (lines 284, 384, 411, 616, and figure 3) There are missing spaces before references throughout the manuscript (lines 177, 181, 214, and 371)
--	--

REVIEWER	Rivera, Windell L University of the Philippines
REVIEW RETURNED	01-Jul-2022

GENERAL COMMENTS	The manuscript submitted by Cremonese et al. describes a protocol for evaluating the simplified sewerage intervention to prevent urban leptospirosis transmission in Salvador, Brazil. A cluster non-randomized trial will be employed to assess the impact of this new system as well as the socio-economic status (SES), topographic or hydrologic characteristics, and population density to the incidence of leptospirosis in highly urbanized low income communities. Overall, the protocol is well-written. The study objectives and expected outputs were presented clearly. However, a few changes should be made to improve the manuscript, specifically:  1. In line 36, include it "is" known. 2. In line 161, a brief background on studies regarding the long term effects of exposure to enteropathogens in child neurodevelopment may be included if the researchers are planning to conduct additional analyses. 3. In line 184, change "depended" on "is dependent". Line 184-185, change "in the short term (5 years-period) to "for a short-term period of 5 years." 4. In line 203, revise the sentence to "Household sewerage networks are interconnected on one main line connecting to the main network, thus...."
--

	5. In line 213, revise the sentence to "...communities which reduced the community participation to sewerage maintenance." 6. Line 220, revise to "...and this study intends to contribute.." 7. In lines 229-231, revise the statement to "we will perform community ethnography to deepen the understanding of the social and cultural impacts caused before and after the intervention, as well as its environmental impacts and the mobilization processes" 8. In line 238, discuss the basis for the increased sample size. How did the authors come up with a total number of 420 additional participants for the other outcomes? 9. In line 253, delete "without" 10. In line 257, delete "far" 11. In line 307, discuss the rationale for the number of environmental samples to be collected. 12. In line 311, add "until molecular assays can be performed." 13. In line 333, how will you ensure the compliance of the residents in using the GPS data logger and/or diary entry? In case of incomplete data due to noncompliance, how will you decide if the data from the participant can still be included in the analysis. Add a withdrawal clause in the protocol. 14. Based on the study by Casanovas-Massana et al., leptospiral load is very low in the environment hence they are very difficult to detect, would the authors want to consider comparing their results to the number of possible reservoir hosts in the area? 15. Given that they will also investigate other communicable diseases, would it be possible to revise the title to include these diseases? 16. If possible, include the questionnaire or case report form that will be used to record the results of parameters mentioned. 17. For enrollment of study participants, kindly mention any measure to ensure data privacy and any confidentiality agreements made between the research team and the community.
--	--

REVIEWER	Simuyandi, Michelo Center for Infectious Disease Research in Zambia
REVIEW RETURNED	19-Dec-2022

GENERAL COMMENTS	The work that this protocol will address is very critical and has a lot of potential for other secondary analysis. I would like authors to also consider screening for arbovirus carriage by vectors as well as serological test (IgM) to also explore prevalence pre- and post intervention.
---

REVIEWER	Barragan, Veronica USFQ, Biology
REVIEW RETURNED	20-Dec-2022

GENERAL COMMENTS	In the protocol entitled "Simplified sewerage to prevent urban leptospirosis transmission: a non-randomized cluster controlled trial protocol in urban disadvantaged communities in Salvador, Brazil", the authors will use an integral approach to evaluate the impact of implementing a simplified and cost effective sewerage system on leptospirosis incidence. This intervention will be implemented in a highly populated and low income urban setting in Salvador, Brazil. The protocol is described in detail and the text justifies the rationale for each of the variables that the researchers will evaluate before and after the intervention. It describes very carefully the phases of the intervention, as well as the methodology that will be used to evaluate the intervention and data analysis.
---

	Four control and intervention clusters will be selected and a baseline for the 8 clusters will be established before intervention. Then, the evaluation of the intervention will be carried out over two and a half years. The protocol is very complete and considers all aspects related to leptospirosis transmission and the impact on the community. It includes detection of new leptospirosis cases, a census of domestic and peridomestic animals, detection of contaminated sources and measurement of leptospira in the environment. It also considers resident movement and includes assessment of community well-being. The number of leptospirosis cases in the intervention clusters is expected to be lower than in the control clusters, and the authors anticipate a 12% decrease in annual cases after intervention. During the time of the intervention evaluation, diarrheal diseases and mosquito-borne febrile illnesses (arboviruses) will also be monitored. Comments:  - The protocol carefully describes how researchers will evaluate the longitudinal impact of the sewerage system by in 4 matched houses clusters. The rationale of cluster matching is explained in a comprehensive manner. - A very valuable aspect of the proposed protocol is that it integrates the community during the intervention process, which will generate community empowerment in the project and in decision making. This is also important because it guarantees the sustainability of the intervention. - The study site is ideal for this intervention and its evaluation since leptospirosis has been closely monitored for several years by the same group of researchers conducting this study. This guarantee the success of the proposed protocol: they have established close links with the community, they know the particularities of human and animal behavior in the community, they have identified local risk factors and important details of the local transmission cycle. - Protocol will allow researchers to record valuable information that could be useful for preventing leptospirosis in other localities of the world with similar conditions. Suggestions: Line 172-273. "Clusters will be matched on socio-economic status (SES), topographic or hydrologic characteristics, population density, and Leptospira seroprevalence." The selection of clusters has been designed to ensure that the control and intervention clusters are comparable. I suggest highlighting that cluster selection will also consider exposure to the pathogen by selecting houses that are located to similar distances to new sewerage system and control open sewers. Figure 1. Inside the Measurements box. Please add in the description of the figure to what HoH stands for. In cluster description, is it possible to add the average size of clusters? In the "STRENGTHS AND LIMITATIONS OF THIS STUDY" section,
--	---

	only strengths are described, would you please describe any limitations of the study?
REVIEWER	Handayani, Farida Dwi University of Nevada Reno School of Medicine
REVIEW RETURNED	27-Dec-2022
GENERAL COMMENTS	This protocol is important and awaited for areas where leptospirosis is a problem in low-income and middle-income countries, as an environmental improvement to prevent leptospirosis. The method used is also sufficient. But it needs to be tightened by repairing rat-proof housed and checking for presence of Leptospira not only in a water and soil but in the kidneys of caught rats. It would also be better to examine the research group who is sick/fever to also do it by PCR or antigen detection, considering that this is a strict cohort study. The fact is that leptospira IgM in the blood will appear very slowly so that it will reduce false negative results. Last but not least, the increase in knowledge, attitude and practice needs to be assessed before and after the intervention.

VERSION 1 – AUTHOR RESPONSE

Referee #1: Dr. Andrea Pires Dos Santos , Purdue University

Comments and suggestions indicated by this referee regarding the restructuring of the text and the writing in English have all been modified in the text and below we answer the main comments:

Comment 1: The authors aim to characterize the mechanisms involved in Leptospirosis transmission, but the role of domestic animals is not explored fully (only census information and no leptospirosis testing). As a suggestion, I believe the inclusion of domestic animal testing (dog/cat) would provide a more one health-focused approach and will add to the characterization of transmission mechanisms of leptospirosis.

Response 1: We are grateful for her positive comments. We consider the inclusion of domestic animal testing (dog/cat) to be important. However, financial support and teamwork are not enough to carry out this component. For this limitation, our research group has applied for new complementary financial grants for the execution of this protocol that foresees the investigation of the role of domestic animals and other animals (e.g., presence of *Leptospira* in the kidneys of caught rats) in the circulation of leptospirosis and other diseases.

Comment 2: My second comment is regarding the current socio-economic situation in Brazil. The authors cite the exclusion of the disadvantaged urban communities' from the government's political agenda; however, the references are from 2003 and 2006. Although this has been a historical

problem in Brazil, I believe a more recent reference would be appropriate. It is also unclear if the authors refer to the three levels of government (federal/state/municipal) or if it is a general statement. Although this is not the focus of the work, a few statements that reflect the current situation would be beneficial to the global understanding of the context of the work. I understand the sanitation company is public and run by the State (Bahia); the work is planned for 2022-2023. Do the authors consider a limitation of the work having no control over the implementation of the sewage system (subjected to possible cancellations or delays, change of power/elections)?

Response 2: Thank you for this comment. We have updated the references in the text that provide evidence about exclusion of the disadvantaged urban communities from the government's political agenda and other information. Please, find the recent references in the lines 86 and 97 of the manuscript.

Regarding the lack of control over the implementation of the sanitary sewage system, we agree that this is an important limitation. However, we have been working with some strategies to try to minimize them. The first is direct institutional contact with a reference person designated by the sanitation company to anticipate possible issues related to the implementation of the sanitary sewage system, ensuring that we can make adjustments to the planning in a timely manner and without major problems to the study. The second refers to the cooperation agreement with the sanitation company. This document contains important commitments for carrying out the depletion and research work within the foreseen period.

Comment 3:

Response 3: We have modified the sentence following reviewer's suggestion. (please see the line 37).

Comment 4: Line 61: In the section "STRENGTHS AND LIMITATIONS OF THIS STUDY," the authors only included the strengths and no limitations (discussed later in the manuscript)

Response 4: We restructured the discussion and modified it according to the suggested comments for a better understanding of the text (please see section "STRENGTHS AND LIMITATIONS OF THIS STUDY").

Comment 5: Line 103: since diarrhea is not a disease per se, I would change it to "diarrheal" diseases or similar.

Response 5: We have modified the sentence following reviewer's suggestion. (please see the line 94).

Comment 6 : Line 111: I would italicize favela throughout the manuscript.

Response 6: We have modified the sentence following reviewer's suggestion.

Comment 7: Line 126: e.g., (missing a comma)

Response 7: We have modified the sentence following reviewer's suggestion.

Comment 8 : Line 228-229: Popular health education actions (no caps)

Response 8: We have modified the sentence following reviewer's suggestion.

Comment 9: Line 370: quantitative polymerase chain reaction (qPCR was used previously)

Response 9: We made modifications in the sentence for clarification (please see the line 306).

Comment 10: Line 393: The word "patient" is not ideal; perhaps participants? Socio-economic and socioeconomic are used in the text; please, standardize them to one format (lines 284, 384, 411, 616, and figure 3). There are missing spaces before references throughout the manuscript (lines 177, 181, 214, and 371)

Response 10: We have modified the sentence following reviewer's suggestion. We modified the text the suggested word and missing spaces.

Referee #2: Prof. Windell L Rivera, University of the Philippines

Comment 1: In line 36, include it "is" known

Response 1: We modified the text (please see the line 37).

Comment 2: In line 161, a brief background on studies regarding the long-term effects of exposure to enteropathogens in child neurodevelopment may be included if the researchers are planning to conduct additional analyses.

Response 2: Thanks to the reviewer for pointing this out. We did a brief background on enteropathogen exposure studies in childhood neurodevelopment following the editor's suggestion (please see the lines 351-361).

Comment 3: In line 184, change "depended" on "is dependent". Line 184-185, change "in the short term (5 years-period) to "for a short-term period of 5 years."

Response 3: We made modifications in the sentence for clarification.

Comment 4: In line 203, revise the sentence to “Household sewerage networks are interconnected on one main line connecting to the main network, thus....”

Response 4: We have modified the sentence following reviewer’s suggestion.

Comment 5: In line 213, revise the sentence to “...communities which reduced the community participation to sewerage maintenance.”

Response 5: We made modifications in the sentence for clarification.

Comment 6: Line 220, revise to “...and this study intends to contribute..”

Response 6: We modified the sentences.

Comment 7: In lines 229-231, revise the statement to “we will perform community ethnography to deepen the understanding of the social and cultural impacts caused before and after the intervention, as well as its environmental impacts and the mobilization processes”

Response 7 : We modified the text. (please see the lines 222/>-224).

Comment 8: In line 238, discuss the basis for the increased sample size. How did the authors come up with a total number of 420 additional participants for the other outcomes?

Response 8: The additional participant numbers in the study considered previous studies in urban communities carried out by the research group. We considered a loss to follow-up of 15-20% per year, so adding 420 participants maintains the appropriate sample size for different outcomes.

Comment 10: In line 253, delete “without” & **Comment 11:** In line 257, delete “far”

Response 10 & 11: We modified the text.

Comment 12: In line 307, discuss the rationale for the number of environmental samples to be collected.

Response 12: We insert a new sentence for clarification.

Comment 13: In line 311, add “until molecular assays can be performed.”

Response 13: We made modifications in the sentence for clarification.

Comment 14: In line 333, how will you ensure the compliance of the residents in using the GPS data logger and/or diary entry? In case of incomplete data due to noncompliance, how will you decide if the data from the participant can still be included in the analysis. Add a withdrawal clause in the protocol.

Response 14: The compliance of the participants will be ensured by the consent form and that they have realized the collection of blood samples. Non-adherence or withdrawal of participants is ensured in every study as listed in lines 446-458. We insert a new sentence for clarification (please see lines 335-338).

Comment 15: Based on the study by Casanovas-Massana et al., leptospiral load is very low in the environment hence they are very difficult to detect, would the authors want to consider comparing their results to the number of possible reservoir hosts in the area?

Response 15: The comment is very pertinent, we intend to compare the results with the abundance and activity of the rodents (reservoir host principal) in the area, according to previous studies by Eyre et al. and Hacker et al.

1. Eyre MT et al. (2020) A multivariate geostatistical framework for combining multiple indices of abundance for disease vectors and reservoirs: a case study of rattiness in a low-income urban Brazilian community *Journal of the Royal Society, Interface* 17:20200398.
2. Max T Eyre et al. (2022) Linking rattiness, geography and environmental degradation to spillover *Leptospira* infections in marginalised urban settings: An eco-epidemiological community-based cohort study in Brazil *eLife* 11:e73120.
3. Hacker KP et al. (2016) A comparative assessment of track plates to quantify fine scale variations in the relative abundance of Norway rats in urban slums *Urban Ecosystems* 19:561–575.

Comment 16: Given that they will also investigate other communicable diseases, would it be possible to revise the title to include these diseases?

Response 16: It will not be possible to change the title to include other communicable diseases. The protocol was designed and has *Leptospirosis* as its main outcome. We're using these efforts to evaluate other diseases, secondarily.

Comment 17: If possible, include the questionnaire or case report form that will be used to record the results of the parameters mentioned.

Response 17: We inserted supplementary material to the main parameters included in the questionnaire. See supplementary table 2.

Comment 18: For enrollment of study participants, kindly mention any measure to ensure data privacy and any confidentiality agreements made between the research team and the community.

Response 18: We insert a new sentence for clarification (please see lines 396-398).

Referee #3: Dr. Michelo Simuyandi, Center for Infectious Disease Research in Zambia

Comment 1: The work that this protocol will address is very critical and has a lot of potential for other secondary analysis. I would like authors to also consider screening for arbovirus carriage by vectors as well as serological test (IgM) to also explore prevalence pre- and post intervention.

Response 1: We are grateful to Reviewer #3 for his positive comments. Our protocol includes screening for arbovirus carriage by vectors with entomological collections, as well as serological test (IgM) to also explore prevalence pre- and post-intervention, and possible change in the abundance and vector diversity.

Referee #4: Dr. Veronica Barragan, USFQ

Comment 1: Line 172-273. "Clusters will be matched on socio-economic status (SES), topographic or hydrologic characteristics, population density, and Leptospira seroprevalence." The selection of clusters has been designed to ensure that the control and intervention clusters are comparable. I suggest highlighting that cluster selection will also consider exposure to the pathogen by selecting houses that are located to similar distances to new sewerage system and control open sewers.

Response 1: We insert a new sentence for clarification (please see lines 165-167).

Comment 2: Figure 1. Inside the Measurements box. Please add in the description of the figure to what HoH stands for.

Response 2: We included the text in the figure legend.

Comment 3: In cluster description, is it possible to add the average size of clusters?

Response 3: We made modifications in the sentence for clarification (please see lines 174-175).

Comment 4: In the "STRENGTHS AND LIMITATIONS OF THIS STUDY" section, only strengths are described, would you please describe any limitations of the study?

Response 4: We made modifications in the sentence for clarification (please see section "STRENGTHS AND LIMITATIONS OF THIS STUDY").

Referee #5: Dr. Farida Dwi Handayani, University of Nevada Reno School of Medicine

Comment 1: This protocol is important and awaited for areas where leptospirosis is a problem in low-income and middle-income countries, as an environmental improvement to prevent leptospirosis. The method used is also sufficient. But it needs to be tightened by repairing rat-proof housed and

checking for presence of *Leptospira* not only in a water and soil but in the kidneys of caught rats. It would also be better to examine the research group who is sick/fever to also do it by PCR or antigen detection, considering that this is a strict cohort study. The fact is that leptospira IgM in the blood will appear very slowly so that it will reduce false negative results. Last but not least, the increase in knowledge, attitude and practice needs to be assessed before and after the intervention.

Response 1: We are grateful for her positive comments. Thanks to the reviewer for pointing this out. This animal component is very important understanding the *Leptospira* prevalence in rats and too in others animals, for example in domestic animals. However, financial support and teamwork are not enough to carry out this component. We intend to address this component in depth over the course of the study with new funding grants. To reduce the limitation, we are collecting data on the presence and abundance of rats and other animals.

Yes, we will be examined the research group who is sick/fever to also do it by PCR and antigen detection. All knowledge, attitude and practice component this protocol will be assessed before and after the intervention in all time.

VERSION 2 – REVIEW

REVIEWER	Dos Santos , Andrea Pires Purdue University, College of Veterinary Medicine
REVIEW RETURNED	03-Feb-2023
GENERAL COMMENTS	bmjopen-2022-065009: The authors have satisfactorily addressed this referee's (#1) comments and suggestions. Thus, my recommendation is to accept the manuscript for publication in BMJ Open.